# Behavioural analysis of factors influencing prescribing for neurodegenerative diseases: A rapid review

Emma Begley[1,2¤], Jason Thomas[1], Carl Senior[1*]

**1** School of Psychology, College of Health and Life Sciences, Aston University, Birmingham, United Kingdom, **2** Alpharmaxim, Altrincham, United Kingdom

¤ Current affiliation: Liverpool City Council
* c.senior@aston.ac.uk

## Abstract

### Background

The incidence and prevalence of neurodegenerative diseases (NDs) are growing worldwide. In an environment where healthcare resources are already stretched, it is important to optimise treatment choice to help alleviate healthcare burden. This rapid review aims to consolidate evidence on factors that influence healthcare professionals (HCPs) to prescribe medication for NDs and map them to theoretical models of behaviour change to identify the behavioural determinants that may support in optimising prescribing.

### Methods and findings

Embase and Ovid MEDLINE were used to identify relevant empirical research studies. Screening, data extraction and quality assessment were carried out by three independent reviewers to ensure consistency. Factors influencing prescribing were mapped to the Theoretical Domains Framework (TDF) and key behavioural determinants were described using the Capability, Opportunity, Motivation – Behaviour (COM-B) model. An initial 3,099 articles were identified, of which 53 were included for data extraction. Fifty-six factors influencing prescribing were identified and categorised into patient, HCP or healthcare system groups, then mapped to TDF and COM-B domains. Prescribing was influenced by capability of HCPs, namely factors mapped to decision making (e.g., patient age or symptom burden) and knowledge (e.g., clinical understanding) behavioural domains. However, most factors were influenced by HCP opportunity, underpinned by factors mapped to social (e.g., prescribing support or culture) and contextual (e.g., lack of resources or medication availability) domains. Less evidence was available on factors influencing the motivation of HCPs, where evident; factors primarily related to HCP belief about

**Data availability statement:** All relevant data are within the manuscript and its supporting information files.

**Funding:** The funding for this work was awarded to CS and JT through an Innovate UK Knowledge Transfer Partnership (KTP) grant (ref 10011439, KTP number 13031) in partnership with Alpharmaxim Healthcare Communications (https://alpharmaxim. com/). KTPs are funded by UK Research and Innovation (UKRI) through Innovate UK and are part of the UK government's Industrial Strategy (https://www.ukri.org/councils/innovate-uk/). Alpharmaxim provided support only for the data extraction stage and reviewing of manuscript drafts; Innovate UK had no role in study design, data collection and analysis, decision to publish, or preparation of the manuscript.

**Competing interests:** I have read the journals policy and the authors of the manuscript have the following competing interests: the research is part funded by Alpharmaxim, where EB has undertaken the KTP-associate placement and therefore has a professional interest. CS and JT are both employed by Aston University and have no competing interests. Our competing interest statement that was noted in the original submission does not alter our adherence to PLOS ONE policies on sharing data and materials.

**Abbreviations:** AChE, acetylcholinesterase; AD, Alzheimer's disease; BPSD, behavioural and psychological symptoms of dementia; CDSS, Clinical Disease Support Service; ChEI, cholinesterase inhibitor; COM-B, Capability, Opportunity, Motivation – Behaviour; DA, dopamine agonist; DMD, disease-modifying drug; DMT, disease-modifying therapy; EPOC, Cochrane Effective Practice and Organisation of Care; GP, general practitioner; HCP, healthcare professional; ICD-9, International Classification of Diseases, ninth revision; L-dopa, levodopa; MDT, Multidisciplinary team; MHRA, Medicines and Healthcare products Regulatory Agency; MMAT, Mixed Methods Appraisal Tool; MS, Multiple sclerosis; ND, Neurodegenerative disease; NHS, National Health Service; NICE, National Institute for Health and Care Excellence; NMDA, N-methyl-D-aspartate; PD, Parkinson's disease; PDU, psychotropic drug use; PRISMA, Preferred Reporting Items for Systematic reviews and Meta-Analysis; QoL, Quality of life; SIGN, Scottish Intercollegiate Guidelines Network; STARR, SelecTing Approaches for Rapid Reviews; TDF, Theoretical Domains Framework

consequences (e.g., side effects) and professional identify (e.g., level of specialism) were often described.

## Conclusions

This systematic analysis of the literature provides an in-depth understanding of the behavioural determinants that may support in optimising prescribing practices (e.g., drug costs or pressure from patients' family members). Understanding these approaches provides an opportunity to identify relevant intervention functions and behaviour change techniques to target the factors that directly influence HCP prescribing behaviour.

---

## Introduction

Neurodegenerative diseases (NDs) are a group of neurological disorders associated with a progressive loss of nerve cells in the brain or nervous system, leading to reduced human functioning [1]. NDs include disorders such as Alzheimer's disease, dementia, Parkinson's disease and multiple sclerosis [1–3]. The most common types of ND are Alzheimer's disease and Parkinson's disease, with the global prevalence of Parkinson's disease doubling between 1990 and 2015, making it the fastest-growing ND [4,5]. The growing incidence and prevalence of NDs are subsequently associated with increasing health and economic burdens, with costs of $51.9 billion for Parkinson's disease in the USA in 2017, $85.4 billion for multiple sclerosis in the USA in 2019 and $1.3 trillion globally for dementia in 2019 [6–8].

At present, there are no cures for NDs and most available medicines only manage disease symptoms [1,2]. However, for healthcare professionals (HCPs), the availability of multiple medications (each with varying efficacies and side-effect profiles) and limited access to patient-specific data have long been identified as obstacles in determining the most suitable treatment [9,10]. While a wide variety of factors can affect treatment choice, it is not known which factors are most important when determining the best treatment for NDs [11].

One way to identify the facilitators and barriers to the optimal management of NDs is to apply a theoretical lens to find key determinates underpinning prescribing behaviour. The Theoretical Domains Framework (TDF) comprises 14 behavioural domains that underpin behaviour; identifying which domains are relevant to a behaviour can aid in understanding what is needed to change that behaviour [12]. Similarly, the Capability, Opportunity, Motivation – Behaviour (COM-B) model, which the TDF domains map onto, features three key components that determine behaviour [13]. This review uses the term 'mapping' to refer to the cross-reference and linkage between two points of data. Together, the TDF and COM-B model can be utilised to enhance the understanding of, and identify ways to change, prescribing behaviour for NDs [11]. For instance, using the TDF, Talet and colleagues [11] conducted a systematic review to identify interventions that could optimise

medication; of the 16 interventions that effectively changed prescribing behaviour, prompts and cues linked to the 'memory attention and decision process' domain of the TDF were most commonly identified, providing a potential target for future interventions.

The objectives of this review involved consolidating evidence from the existing literature to 1) comprehensively identify factors influencing prescribing for NDs and 2) map these factors to the TDF and the COM-B model to understand the theoretical determinants underpinning prescribing practices. As is noted below, the initial aim was to focus specifically on Parkinson's disease, but this revealed limited citations, thus the search was subsequently broadened to include all NDs.

## Methods

The protocol and methods chosen for this rapid review were informed by recent guidance for performing rapid reviews [14]. To accommodate the rapid timeline, the SelecTing Approaches for Rapid Reviews (STARR) decision tool was used to support decisions, including whether to limit the scope of the review and associated search strategy (described below); report a narrative synthesis of included studies; search two open access database sources; and share screening, data extraction and quality assessment between two reviewers, with a third reviewer to resolve any disagreements [15]. The Preferred Reporting Items for Systematic reviews and Meta-Analyses (PRISMA) checklist for scoping reviews was used to ensure review transparency [16], and a protocol was also developed using the PRISMA protocols checklist [17] and revised by the research team. This was registered with PROSPERO on 13/09/2023, registration number CRD42023461578, accessible at https://www.crd.york.ac.uk/PROSPERO/display_record.php?RecordID=461578. Ethical approval was not sought for this study as it only reviews existing published literature that have previously obtained their own approvals.

### Study inclusion criteria

Empirical research studies were considered for inclusion if they met all three of the following criteria:

1. The primary outcome reported factors that influenced decisions about prescribing (i.e., when to initiate, titrate or switch treatments) for a ND.

2. Studies utilised qualitative and/or quantitative methodology.

3. Journal articles were peer reviewed, published in English language and involved human participants.

Articles were excluded if they reported on non-drug prescribing (e.g., physical therapy) or explicitly explored patient or carer perspectives about prescribing. Any systematic reviews were screened for secondary references. See S1 Appendix for the full inclusion and exclusion criteria.

### Information sources and search strategy

Two literature databases, Embase and Ovid MEDLINE, were chosen due to their wide disciplinary focus on the research topic. All relevant articles from database inception to 1 March 2024 were retrieved for screening. A limited scoping search was initially carried out on Ovid MEDLINE, using only keywords that addressed the research question. On review of this search, a more comprehensive search strategy was developed by the lead author, further refined by an expert librarian and refined by subsequent discussions with the research team. The initial aim was to target Parkinson's disease specifically; however, the search strategy revealed few citations (<123) and, therefore, it was decided to broaden the search to include any ND. The final search strategy for Ovid MEDLINE can be found in S2 Appendix. A syntax of keywords and Medical Subject Headings (MeSH) was individually entered into each database and wildcards or truncations were used to find variations in spellings or word endings. The terms targeted the following categories:

- Population: clinician, consultant, doctor, general practitioner (GP), NDs, Parkinsonian disorders.

- Intervention: drug prescription.

- Outcome: prescribing, decision, choice.

### Screening and study selection

Search results were exported into EndNote, where a robust 13-step deduplication process was performed that combined variations of author, year, title or journal details. The remaining articles were divided between two reviewers for screening of titles, abstracts and full-text articles, applying the inclusion/exclusion criteria outline in S1 Appendix. The lead author screened 10% of the second reviewer's articles to ensure consistency and any disagreements were resolved via consensus with a third reviewer.

### Data extraction and quality assessment

Items from the Cochrane Effective Practice and Organisation of Care (EPOC) data collection checklist were used to guide data extraction. Extracted data included study methodology (e.g., design, aim), participant and disease characteristics (e.g., demographic information, neurological condition), outcomes measured (e.g., prescription decision) and study results (e.g., prescribing determinants). The most recent version of the Mixed Methods Appraisal Tool (MMAT) was used to critically assess the quality of the included studies [18]. A quality assessment score and description for each study design is reported. All relevant data are contained within the manuscript and its supporting information files and, therefore, are freely available.

### Data synthesis

A synthesis of the evidence is narratively reported for each study design. Factors found to influence prescribing decisions were first mapped onto the TDF to identify which domains of behaviour change should be addressed when designing interventions to optimise medication choice [12]. The 14 domains were knowledge; skills; social/professional role and identity; beliefs about capabilities; optimism; beliefs about consequences; reinforcement; intentions; goals; memory, attention and decision making; environmental context and resources; social influence; emotion; and behavioural regulation [12].

A codebook was developed for mapping factors and behavioural constructs and is available in S3 Appendix. Relevant TDF domains were then cross-referenced to the COM-B components to identify and understand the behavioural determinants that need to be targeted to support optimal prescribing for NDs. Factors mapped to each COM-B domain were then arranged into patient, HCP or healthcare system groups.

## Findings

### Study selection

A total of 3,099 articles were retrieved from database searches ran in December 2022. Records were commonly excluded at title and abstract level for being conference abstracts or duplications (n=1,758). Of the 187 full-text articles reviewed, most were excluded because they did not report on factors influencing clinician prescribing decisions (n=55); reasons for further exclusion are reported in Fig 1. Data were extracted for 53 articles. An updated search in March 2024 yielded 124 additional articles, of which only one was included for data extraction; the processing of these articles is reported in brackets in Fig 1. Inclusive of the updated search, a total of 54 articles met the inclusion criteria and were selected for data extraction.

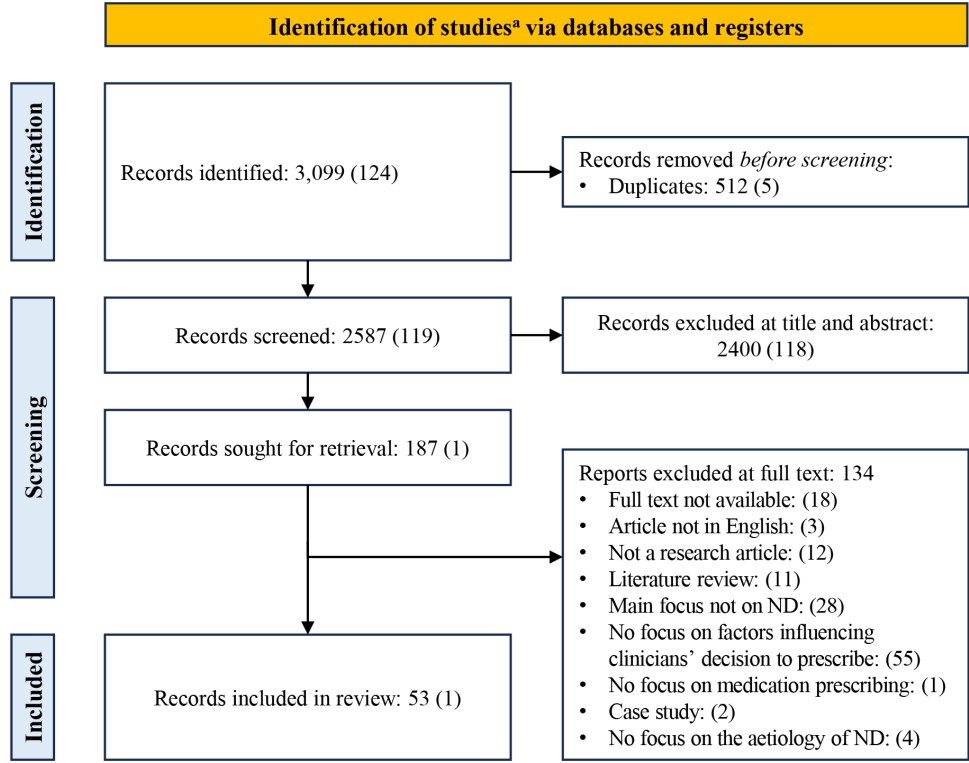

**Fig 1. PRISMA diagram of article screening and selection.** [a] The number of records at each stage are presented; the updated search records are presented in brackets. ND, neurodegenerative disease.

## Study characteristics

Of the 54 studies included, 38 were quantitative (e.g., survey or database analysis), 12 were qualitative (e.g., interviews or focus groups) and four were mixed methods studies (e.g., interview, case note and survey analysis). The sources of data used to explore prescribing practices comprise a variety of HCP perspectives using qualitative, quantitative or mixed methods and prescription claims or patient linkage databases. A description of each study and the factors influencing prescribing are summarised in Table 1. The included studies focus on prescribing for dementia (n=25), followed by Parkinson's disease (n=16), Alzheimer's disease (n=8) and multiple sclerosis (n=5). Most studies sought to explicitly understand HCP prescribing decisions; however, insight about prescribing could also be identified from studies that were designed to explore prescribing trends, intervention evaluations, medication management and HCP knowledge or opinions of pharmacological treatment.

## Quality assessment of included studies

More than half of the studies included (n=31) were rated high quality (scoring 5*) (see S4 Appendix for full assessment scores). One study (mixed methods) did not meet any quality criteria. It was difficult to assess quality for 14 non-randomised, four descriptive and two mixed methods studies due to them not indicating complete outcome datasets, any confounders or risk of bias.

## Synthesis of prescribing determinants

Fifty-six factors were identified that influenced prescribing decisions. Symptom burden was reported by most studies (n=21), followed by patient age (n=20) and side effects (n=16). Treatment preference, HCP specialism, drug safety/

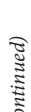

**Table 1. Study characteristics.**

| Author [year]; country | Aim | Design; source of data | Population [sample size]; setting | Clinical disease [intervention] | Outcome(s) measured | Results | Prescribing determinants | Quality score |
|---|---|---|---|---|---|---|---|---|
| Bell et al., [2020]; USA [19] | Evaluate an intervention designed to improve prescribing in a veterans nursing facility. | Evaluation; national prescribing database and online questionnaire. | Clinical providers [N=7]; skilled nursing facility. | Dementia [psychotropic medication]. | Number of new patients requiring medication review, uptake of pharmacist recommendations. | Patients identified for medicine changes (n=9), resulting in uptake of 66% of recommendations. | Pharmacist recommendations, symptom burden, family preference, prior patient bad experience. | 4* |
| Cameron et al., [2019]; UK [20] | Identify factors influencing prescribing DMT. | Qualitative design; semi-structured interviews. | Consultant neurologists [N=18], specialist nurses [N=16]; 15 NHS sites. | MS [DMT]. | Views and experiences of clinicians who prescribe DMT. | Five themes influencing prescribing emerged (listed as determinants in next column). | Prescribing guidelines, identifying relapses, perceived risk, readiness to prescribe, familiarity and prior experience and peer networks and prescribing culture. | 5* |
| Cousins et al., [2017]; Australia, Tasmania [21] | Identify factors influencing prescribing psychotropic medication. | Survey; GPs. | GPs [N=177]; nursing homes. | Dementia [psychotropic medication]. | Factors reducing prescribing psychotropic medication, prescribing habits, expectation of medication benefit. | Need to target staffing levels and resources to reduce prescribing psychotropic medication in nursing homes. | Other HCPs (nurses/pharmacists), family members, HCP experience, pressure to prescribe due to lack of non-pharmacological options, risk to QoL, HCP training, GP confidence. | 5* |
| Crispo et al., [2015]; USA [22] | Describe patterns and trends of anti-PD drug prescribing in the USA between 2001 and 2012. | Retrospective cohort analysis; Cerner Health Facts database. | Individuals >40 years of age with PD diagnosis ICD-9 code 332 [N=16,785]; inpatient records. | PD [anti-PD drugs]. | Annual prevalence of anti-PD drug use by patient age, sex, race and census region and factors affecting prescribing practice. | L-dopa was most prescribed. Proceeding regulatory safety concerns, dopamine agonist prescriptions decreased except in patients >80 years of age. | Safety concerns, cognitive impairment, age, gender. | 5* |
| Cross et al., [2020]; Australia [23] | Explore barriers and enablers of stakeholder roles in medication management. | Qualitative design; focus groups. | Consumers, GPs, nurses, pharmacists [N=55]; primary, tertiary, residential aged care. | Dementia [medication management]. | Views and experiences of perceived role in managing medication for dementia. | Data from nine focus groups were analysed, from which four main themes emerged: supporting the role of the person with dementia; carer roles and challenges; HCP roles; and process and structure barriers to medication management. | Patient cognitive and functional impairment, patient/carer preference, changes to medication regimen, GP assessment, MDT communication/ support, therapy goal, side effects, comorbidity, polypharmacy, carer burden, medication reviews, HCP fear of consequences, treatment evidence, guidelines, clinician training. | 5* |

*(Continued)*

| Author [year]; country | Aim | Design; source of data | Population [sample size]; setting | Clinical disease [intervention] | Outcome(s) measured | Results | Prescribing determinants | Quality score |
|---|---|---|---|---|---|---|---|---|
| Degli Esposti et al., [2016]; Italy [24] | Explore prescribing pattern and resource use with patients with PD prescribed rasagiline or selegiline. | Retrospective cohort study; prescription, hospital discharge, ambulatory care databases. | Patients with PD [N=1,607]; prescription for MAO-B inhibitor. | PD [rasagiline or selegiline]. | Anti-PD, drugs, other MAO-B inhibitor prescriptions, hospitalisation for PD. | Increased selegiline prescriptions; increased hospitalisations and anti-PD prescriptions in rasagiline group. | Patient age and gender, comorbidity, hospitalisations, polypharmacy. | 5* |
| Degli Esposti et al., [2017]; Italy [25] | To describe changes and predictors of change in DMT for MS. | Observational cohort study; prescription databases. | Patients with MS prescribed first line injectable DMT [N=1,698]; local health authority data. | MS [DMT] | Changes in therapy due to lack of adherence, therapeutic switch, temporary discontinuation or interruption. | Increased age effected non adherence, switch and temporary discontinuation. Likelihood of switch decreased with length of disease. Females or naive patients more likely to temporarily discontinue treatment. | Non adherence, age, length of disease, | 4* |
| Desai et al., [2019]; USA [26] | Describe time trends and identify factors with initiating and switching oral DMD. | Retrospective cohort study; healthcare usage claims database. | Patients with MS [N=8,918]; receiving DMD. | MS [DMD]. | Changes in DMD prescriptions over time, predictors of initiating or switching treatment. | Patients initiating or switching oral DMD increased between 2011 and 2014, driven by patient and physician preferences, formulary status and insurance coverage. | Patient age, symptomatic treatment, side effects, polypharmacy, neurologist consultation, emergency room visit, geographical location. | 5* |
| Dhuny et al., [2021]; Ireland [27] | Explore knowledge, attitudes and opinions of GPs prescribing psychoactive drugs for dementia. | Descriptive cross-sectional; survey. | GPs [N=168]; primary care. | Dementia [psychoactive drugs]. | Association between prescribing and GPs' years of experience in primary care, location of practice, ease of referral. | Consensus that antipsychotics did not benefit all patients and there is a need for more GP medication management training. | Pressure from nurses, severity of patient aggression, impact of withdrawing medication on patient QoL and return of symptoms, confidence to manage medication, access to specialists. | 5* |
| Disalvo et al., [2020]; Australia [28] | Explore medication-related decision-making for dementia. | Qualitative design; focus groups. | HCPs [N=16 HCPs in N=4 focus groups]; patients in long-term care. | Dementia [antibiotics, lipid-lowering agents, opioids, AChE inhibitors]. | Decisions to start, continue or deprescribe medication. | Need to consider a person-centred approach and collaborative MDT input to reach quality medication decision-making. | Goals of care, MDT, family and patient input, patient variability, disease trajectory, drug risk versus benefit, polypharmacy, comorbidity, cardiovascular risk, medication history, adverse effects, drug interactions, severity of symptoms, QoL, swallowing capacity, incontinence level, HCP inertia. | 5* |

*(Continued)*

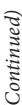

| Author [year]; country | Aim | Design; source of data | Population [sample size]; setting | Clinical disease [intervention] | Outcome(s) measured | Results | Prescribing determinants | Quality score |
|---|---|---|---|---|---|---|---|---|
| Donyai, [2017]; UK [29] | Explore HCP deliberations about antipsychotic medication prescribing for dementia. | Qualitative design; semi-structured interviews. | HCPs, care home managers [N=28]; care homes. | Dementia [antipsychotic medication]. | Explore the use of fallacious arguments for prescribing antipsychotic medication in care homes. | Fallacious arguments were formulated to justify prescribing behaviours for antipsychotic medication. | False dichotomy, medication popularity, tradition, consequence, emotion or fear of prescribing, disruption of care due to prescribing versus not prescribing, care home management. | 5* |
| Duthie and Banerjee, [2011]; UK [30] | Examine the dissonance of guideline messages on prescribing antidementia medication. | Mixed methods; case notes and survey. | Old age psychiatrics [N=378 case notes and N=57 survey responses]; hospitals. | Dementia [ChEIs]. | Practice and attitudes of routine prescribing in relation to national guidelines. | Clinicians more likely to adhere to local prescribing protocol, however, often adhere to SIGN guidelines more so than NICE. | Guidelines, diagnosis, mental health score, clinical experience and judgement. | 0* |
| Earla et al., [2020]; USA [31] | Examine prescribing patterns and trends of DMT. | Cross-sectional design; national level survey data from 2006 to 2015. | Patients with MS visits [N=3.84 million] resulting in a prescription; office-based outpatient visits. | MS [DMT]. | Descriptive prescribing patterns, factors associated with DMT prescriptions. | Between 2010 and 2015, oral DMTs increased; between 2006 and 2016, injectable DMTs decreased. Predisposing and enabling factors influenced prescribing. | Age, region, physician specialty, visits to neurologists. | 5* |
| Fargel et al., [2007]; USA, Europe [32] | Evaluation of treatment options, satisfaction and opinions about treatment improvements for PD. | Cross-sectional; survey data. | Patients with PD [N=500], neurologists [N=592]; treatment centre/neurologist practice. | PD [anti-PD medication]. | Assessment and classification of current treatment options, satisfaction of treatment, prescribing trends, desired improvements in medication. | Efficacy and safety are important for any medication improvements. More research on treatment options is needed to alleviate symptoms, improve adherence, drug administration and guidelines regarding aspects for QoL. | Patient preference, stage of disease, adverse effects, tolerability, reduced dosing, efficacy, cost, improved QoL. | 4* |
| Gardette et al., [2014]; Europe [33] | Investigate incidence and predictors of switching and discontinuing ChEIs. | Prospective cohort study; questionnaire. | Patients with AD [N=557]; memory clinics. | AD [ChEI medication]. | Reasons for changing or withdrawing ChEIs and choice of new therapy. | The incidence of switching was higher than discontinuation; key reasons were drug inefficacy and adverse drug events, respectively. | Discontinuation: adverse events, centre speciality, patient/caregiver preference, compliance, inefficacy, drug cost, cardiac comorbidity, hospitalisation, falls, reduced mental health scores. Switch: adverse events, polypharmacy, inefficacy, drug cost, disease severity, high nurse resource, worsening symptoms. | 4* |

*(Continued)*

**Table 1.** (Continued)

| Author [year]; country | Aim | Design; source of data | Population [sample size]; setting | Clinical disease [intervention] | Outcome(s) measured | Results | Prescribing determinants | Quality score |
|---|---|---|---|---|---|---|---|---|
| Gill et al., [2019]; UK [34] | Explore professionals' social construction of using antipsychotics for dementia. | Qualitative; semi-structured interviews. | HCPs, care home managers [N=28]; dementia care homes. | Dementia [antipsychotics]. | Professionals' social constructions and realities about prescribing antipsychotics in dementia. | Prescribing was justified as being *'the lesser of two evils'*, however, awareness of overprescribing was also prominent. | An aid to patients and carers, fast delivery of care, reduced risk of disease-related harm, QoL, carer and HCP preference, adverse effects, demand on care home staff, ease and convenience of prescribing. | 5* |
| Grandas and Kulisevsky [2003]; Spain [35] | Assess patterns of PD drug use in Spain. | Population-based study; nationwide survey. | Surveyed physicians [N=241] on treatment of N=1,803 patients; various levels of healthcare. | PD [anti-PD drugs]. | Clinical patient characteristics and drug type and class prescribed by physician. | L-dopa was the most prescribed drug, regardless of clinician. Prescriptions for DA, catechol-o-methyltransferase inhibitors and anticholinergics were influenced by clinician specialism. | Specialism of clinician. | 3* |
| Green et al., [2019]; USA [36] | Investigate clinician-perceived barriers and facilitators for optimised prescribing for dementia. | Qualitative; semi-structured interviews. | HCPs [N=21]; primary and specialist care. | Dementia [prescription medication]. | Factors perceived to facilitate or create barriers to optimised prescribing. | Prescribing medication for dementia is complex. Factors influencing optimised prescribing are described by: 1) clinician-perceived barriers; 2) perceptions of enablers; and 3) approaches to discussing medications. | MDT and service access, guidelines, patient and caregiver beliefs and expectations, caregiver availability and skills, lack of data on efficacy and safety, uncertainty of adverse effects or benefits, caregiver burden, comorbidity, polypharmacy, practicality of and access to non-pharmacological resources, clinician confidence, knowledge, specialisation, cognitive bias, therapeutic inertia, lack of time. | 5* |
| Hanson et al., [2014]; USA [37] | Understand neurologists' decision-making and experience of prescribing DMT for MS. | Cross sectional; online survey design. | Neurologists or MS specialists [N=102]; specialist care. | MS [DMT]. | Perceived patient adherence, satisfaction, frequency of patient contact and reason for contact. | Majority of patients are adherent and satisfied with DMT. Calls to neurologists are mostly related to side effects or insurance coverage. Decisions to prescribe are presented in order of importance in the next column. | Efficacy, safety, tolerability, patient preference, convenience; frequency of relapse, worsening disease progression related to switching. | 3* |

*(Continued)*

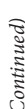

| Author [year]; country | Aim | Design; source of data | Population [sample size]; setting | Clinical disease [intervention] | Outcome(s) measured | Results | Prescribing determinants | Quality score |
|---|---|---|---|---|---|---|---|---|
| Houghton et al., [2019]; USA [38] | To understand standard of care treatment for PD patients and factors associated with treatment choice. | Retrospective cohort analysis; Medicare claims data. | Patients with PD [N=68,532]; in/out patient hospital setting. | Parkinsons [APD's]. | APD, time to initiation, time of treatment switch/add on. | Duration to first prescription since diagnosis, 37 days; Ldopa most commonly prescribed; age correlated with treatment initiation, drug choice and likelihood of switch | Age, gender, comorbidity | 5* |
| Hillmer et al., [2006]; Canada [39] | Describe family physicians' prescribing practices for AD. | Cross sectional; survey design. | Family physicians [N=334]; practice. | AD [ChEIs]. | Frequency of prescribing ChEIs, perceived effectiveness, appropriate time to prescribe, factors influencing prescribing, clinician gender, self-reported clinical knowledge, formulary coverage, visits from pharma, clinical practice information. | ChEI prescribing is higher in regions where it is available with formulary coverage. Factors influencing prescribing are reported in the next column. | Formulary coverage, female HCPs, perception of effectiveness, clinician knowledge of ChEIs. | 4* |
| Hoffmann et al., [2011]; Germany [40] | Investigate the influence of geriatric comorbidity and polypharmacy on ChEI prescriptions for dementia. | Cohort study; insurance claim database. | Patients with dementia [N=1,848]; outpatient care. | Dementia [ChEIs]. | ChEI prescription, geriatric comorbidity, polypharmacy. | Patients receiving level 3 care were unlikely to receive a prescription, however, this decreased with more geriatric symptoms. Patient age was independently significant; locality and contact with specialists were linked to greater likelihood of a prescription. | Level of care, symptoms of geriatric complexes, patients' age, rural living, contact with neurologists or psychiatrists. | 5* |
| Jani and Prettyman [2001]; UK [41] | Monitor compliance with protocol for prescribing donespezil to treat AD. | Prospective clinical audit; patient clinical assessment data and carer questionnaires. | Patients with AD receiving donepezil [N=35]; hospital. | AD [donepezil and ChEIs]. | Dementia severity and outcome of therapy by cognitive and functional performance, carers expectations and knowledge of therapy. | Likelihood of receiving a prescription was determined by patient characteristics consistent with eligibility in the protocol. | Protocol guidance, failure to meet diagnostic criteria, HCP refusal to prescribe, follow-up issues, discontinuation linked to side effects, worsening of disease, lack of benefit, resource and organisational factors. | 2* |

*(Continued)*

| Author [year]; country | Aim | Design; source of data | Population [sample size]; setting | Clinical disease [intervention] | Outcome(s) measured | Results | Prescribing determinants | Quality score |
|---|---|---|---|---|---|---|---|---|
| Jeschke et al., [2011]; Germany [42] | Analyse prescribing patterns in HCPs specialised in complementary and alternative medicine and compared with current guidelines. | Prospective 5-year multi-centre study; observational study using secondary data. | Physicians [N=22]; treating patients with dementia in primary care. | Dementia [antidementia drugs]. | Factors associated with the prescription of any antidementia drug and *Ginkgo biloba*: patient age, gender, comorbidity, physician age, gender specialisation, year of prescription, type of prescription, type of dementia, polypharmacy. | A prescription for any antidementia drug was >1 when patients had seen a neurologist, had an AD diagnosis, comorbidities of hypertension, heart failure and a neuroleptic therapy. Frequency of antidementia prescribing is equivalent across Germany; however, *Ginkgo biloba* is higher. | Any antidementia drug: HCP specialism, diagnosis of AD, comorbidities (hypertension or heart failure). *Ginko biloba*: patient gender and age, HCP specialism, type of dementia. | 5* |
| Kerns et al., [2018]; USA [43] | Evaluate primary care prescribing strategies for BPSD. | Qualitative; semi-structured interviews. | Physicians [N=26]; primary care. | Dementia [non-pharmacologic and drugs]. | Non- pharmacologic and drug strategies used to treat BPSD. | BPSD management was described across four themes: 1) barriers to non-pharmacologic methods; 2) medication is easy, effective, reasonably safe and appropriate; 3) impact of pharmacologic policies; and 4) evidence-based guidelines to manage BPSD. | Patient therapy goals and needs, diagnosis, efficacy, risk of harm, evidence base (including lack of guidelines) and clinician training, ease of prescribing, affordability, availability, quick to take effect, side effects, physician experience, QoL, level of specialism, carer and professional preference. | 5* |
| Martin et al., [1999]; UK [44] | Determine health authority advice and clinician attitudes of prescribing donepezil. | Cross sectional; postal survey data. | Pharmaceutical advisors [N=75], GPs [N=795]; primary care. | AD [donepezil]. | Knowledge, investigation and management of AD; opinions about diagnosis and treatment; background practice and demographic characteristics, health authority prescribing advice. | Compared with non-prescribers, early prescribers of donepezil agreed that new drugs should be prescribed for mild and moderate AD, that they should be initiated by a GP and that cost should not be an issue. | Severity of disease, male prescriber, health authority advice, role of HCP, carer input in determining drug effectiveness. | 3* |
| Martinez-Lage [2010]; France, Germany, Italy, Spain, UK [45] | Assess physician attitudes and perceptions of treatment and diagnosis, carers and family of patients and government responses to AD. | Cross sectional; survey data. | Generalists [N=250], specialists in AD [N=250]; healthcare. | AD [clinical healthcare for AD]. | Personal attitude, behaviours, and perceptions of ageing and the elderly, signs and symptoms of AD, diagnostic concerns, treatment issues, social impact of AD and role of the government. | Most physicians believe treatment for AD has improved, however, they report that the disease is still undertreated. | Severity of disease, under or delayed diagnosis, efficacy, cost, government restrictions, worsening symptoms, delay progression. | 3* |

*(Continued)*

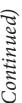

| Author [year]; country | Aim | Design; source of data | Population [sample size]; setting | Clinical disease [intervention] | Outcome(s) measured | Results | Prescribing determinants | Quality score |
|---|---|---|---|---|---|---|---|---|
| McIlroy et al., [2015]; UK [46] | Investigate changes in antipsychotic prescribing for dementia following safety warnings. | Time series analysis; electronic patient records. | Admitted patients with dementia [N=1,085]; hospital. | Dementia [second-generation antipsychotics]. | Demographic and clinical patient data, monthly rates of antipsychotic prescription before and after safety warning. | Drug warnings influenced a reduction in prescribing. Lack of alternative options may explain continued use. | Drug safety warnings, off-label prescribing, age, longer hospital stays, polypharmacy. | 5* |
| Monette et al., [2012]; Canada [47] | Evaluate an educational intervention on antipsychotic prescribing for dementia. | Longitudinal study; care home database and resident chart review. | Residents [N=429], members of MDT [N=372]; long-term dementia care home. | Dementia [educational intervention targeting antipsychotic drug prescribing]. | Pre-, during and post-evaluation of changes in the odds of prescribing antipsychotics. | Educational interventions in long-term care may support a reduction in antipsychotic prescribing in the short term. | HCP education (including consciousness raising, educational sessions and clinical monitoring). | 5* |
| Neo et al., [2020]; Singapore [48] | Compare PD prescribing among specialists and identify factors that influence drug choice. | Prospective temporal analysis; national movement disorder database. | Patients with PD [N=230]; seen at a national neuroscience institute. | PD [PD drugs]. | Choice of PD drugs and changes in prescribing patterns. | In the last decade, choice of drugs for newly diagnosed patients with PD has changed; influenced by patient age and stage of disease. Emerging evidence and reports of adverse drug effects may have contributed to changes in use. | Stages of disease, symptom severity, patient age. | 5* |
| Nijhuis et al., [2016]; Netherlands [49] | To explore decision making in advanced PD | Qualitative study; focus group and 1:1 interview data. | Patients with PD [N=20]; caregivers [N=16]; neurologists [N=7]; PD nurses [N=3]. Hospital setting. | PD [deep brain stimulation, Ldopa gel or apomorphine infusion] | Factors influencing shared decision making | Shared decision making centred around 4 key themes: 1) information needs, 2) factors influencing choice, 3) decision making roles and 4) barriers and facilitators of decision making. Factors influencing HCPs choice are reported in next column. | Professional preference/ experience, logistics of aftercare, patient characteristics (e.g., support, capability, expectations, burden), severity of symptoms, comorbidity, treatment evidence, guidelines, treatment characteristics and contra indications, cost, availability. | 5* |
| Ooba [2011]; Japan [50] | Assess the impact of regulatory action alerting the risks with prescribing dopamine receptor agonists. | Retrospective prescription analysis; Japanese medical claims database. | Patients with PD >40 years of age [L-dopa N=620; ergot DA N=211; non-ergot DA N=214, anticholinergic agents N=434]; inpatient and community. | PD [dopamine receptor agonists]. | Changes in patients prescribed cabergoline or pergolide before and after regulatory action and rate of ultrasonic cardiography examination. | Regulatory action did not significantly reduce cabergoline or pergolide prescriptions, but increased ultrasonic cardiography examination. | Regulatory did not have the desired effect and no other prescribing determinants were reported. | 3* |

*(Continued)*

**Table 1.** (Continued)

| Author [year]; country | Aim | Design; source of data | Population [sample size]; setting | Clinical disease [intervention] | Outcome(s) measured | Results | Prescribing determinants | Quality score |
|---|---|---|---|---|---|---|---|---|
| Orayj et al., [2021]; Wales [51] | Investigate changes in and predictors of treatment choice for PD. | Retrospective cohort study; a national data linkage databank that includes GP prescribing. | Newly diagnosed patients with PD [N=9,142]; population healthcare. | PD [PD drugs]. | Predictors of PD medication prescribing. | There is a significant switch toward prescribing L-dopa in Wales, providing some indication of compliance with guidelines on safety and efficacy. | Age, gender, social deprivation, comorbidity (diabetes), antidepressants. | 4* |
| Oremus et al., [2007]; Canada [52] | Provide examples of clinical efficacy measures in AD and their impact on prescribing. | Descriptive design: postal survey. | Geriatricians, neurologists, psychogeriatricians, GPs [N=233]; clinical care. | AD [scenario of a hypothetical new medication]. | Efficacy requirements of prescribing a new hypothetical new medication in place of ChEIs. | Physicians would consider prescribing a new drug if it allowed patients to remain in a mild/moderate disease state; however, halting or reversing disease progression was a preferred prerequisite for most physicians. | Stage of disease, halting disease progression, prevention of worsening symptoms. | 5* |
| Ott et al., [2023]; USA [53] | Describe and compare prescribing patterns during 2011–2018 and identify recipient characteristics for antidementia drugs before and after nursing home admission. | Retrospective observational study; Medicare database data. | Nursing home residents [350,197]; Nursing Home (NH). | Dementia [ChEIs and memantine medication]. | Prevalence and resident characteristics pre/post NH admission for initiating, discontinuing and continuous use of ChEIs and memantine medication. | ChEIs and memantine declined pre-NH admission between 2011–2018. Initiation was lower in older residents, using a feeding tube and greater functional dependency; discontinuation was higher in residents with comorbidities, aggressive behaviour, function dependency, cognitive impairment and hospital stays. | Age, Alzheimer disease diagnosis, use of a feeding tube, greater functional dependency, comorbidity, cognitive impairment, aggressive behaviour, hospital stays. | 5* |
| Peisah et al., [2015]; Australia [54] | Identify components of quality prescribing in BPSD; develop and test a tool for quality prescribing. | Mixed methods design; Delphi consensus and tool evaluation. | HCPs for the Delphi study [N=12], HCPs who tested the tool [N=48]; inpatient and ambulatory. | Dementia [psychotropics]. | Components of quality prescribing; development and efficacy test of an education tool for quality prescribing. | The new tool supported improve quality prescribing for dementia. Factors related to quality prescribing are reported in the next column. | Failure to use non-pharmacological alternatives, indication for drug, evidence of efficacy, patient consent, mode of administration, polypharmacy, side effects and review. | 5* |
| Petrazzuoli et al., [2020]; 25 European countries [55] | Explore primary care management for dementia. | Mixed methods, cross-sectional design; survey questionnaire. | Physicians [N=445]; primary care settings across 25 European countries. | Dementia [drug treatment for patients with dementia]. | Processes and typologies of dementia management and proportion of case studies that adhere to drug treatment guidelines. | Unburdening dementia was a focus of dementia management and targeted by recognising and assessing the burden. 60% of prescribing was in accordance with guidelines. | Relieving disease burden, availability of drugs, geographical guidelines, country prescribing regulations, diagnosis, level of HCP specialism. | 5* |

Table 1. (Continued)

| Author [year]; country | Aim | Design; source of data | Population [sample size]; setting | Clinical disease [intervention] | Outcome(s) measured | Results | Prescribing determinants | Quality score |
|---|---|---|---|---|---|---|---|---|
| Podhorna et al., [2020]; France, Germany, Japan, UK, USA [56] | Exploration of real-world physician behaviour in the treatment of AD. | Cross sectional; questionnaire and patient record data. | GPs, neurologists, geriatricians, psychiatrists [N=1,086]; primary and secondary care. | AD [AD-specific pharmacologic treatment]. | Factors influencing any AD-specific prescribing, and factors influencing initiation of AD treatment. | Physicians were unlikely to prescribe due to patient refusal and early-stage AD. Attitudes may be influenced by limited awareness of benefit of early intervention and modest efficacy of treatment. | Patient refusal, stage and severity of disease, symptom control, cognitive status, QoL, tolerability, adherence, pill burden, monitoring, medication, caregiver support, physician knowledge and belief of efficacy, physician type. | 4* |
| Rochon et al., [2018]; Canada [57] | Explore how male and female physicians differ in their prescribing practice of ChEI drugs for dementia. | Retrospective cohort study; healthcare databases. | Clinical physicians [N=9,254]; treating patients with dementia. | Dementia [ChEI drugs]. | Assessment of prescribing practices by physician gender on their initiation of low-dose ChEI drugs. | Female physicians were more likely than males to initiate ChEI drugs at a lower dose, for a shorter duration and follow up with cardiac screening. Initial dosing also differed by physician speciality. | Physician gender and speciality. | 5* |
| Schröder et al., [2011]; Germany [58] | Investigate adherence to PD guidelines in Germany. | Cross sectional; patient record survey. | Neurologists [N=60]; outpatient and community. | PD [anti-PD drugs]. | Assessment of guideline adherence for prescribing DA and L-dopa by patient age and grade of functional impairment. | Prescribing guidelines were moderately adhered to in patients <70 years of age (53% and 84%) and >70 years of age (50% and 52%) with lower and higher functional impairment, respectively. | Functional impairment, age, symptom severity, time since diagnosis. | 4* |
| Smeets et al., [2014]; Netherlands [59] | Explore factors related to psychotropic drug prescribing for dementia. | Qualitative; semi-structured interviews | Physicians [N=15], nurses [N=14]; treating patients in nursing homes. | Dementia [psychotropic drugs]. | Factors related to prescribing decisions. | Factors influencing psychotropic drug prescribing were grouped into four themes: 1) mindset; 2) knowledge and experience; 3) communication and collaboration; and 4) external factors. | HCP beliefs, knowledge (efficacy, side effects, guidelines) and experience of prescribing. Joint decision-making, limited staff/skills, nursing home setting, access to consultants, prescribing policies. | 5* |
| Stephens et al., [2014]; UK [60] | Describe changes in and impact of clinical and socio-demographic factors on antipsychotic prescribing for dementia between 2010 and 2012. | Retrospective longitudinal design; inpatient hospital database. | Patients with dementia [N=10,440]; 34 hospital settings. | Dementia [antipsychotic prescribing]. | Socio-demographic and clinical factors impacting changes in prescribing. | Antipsychotic prescribing fell and was influenced by patient socio-demographic and clinical factors. | Inpatient hospital care, diagnosis of schizophrenia or dementia, emotional state of patient, comorbidity, deprivation, gender, age. | 4* |

*(Continued)*

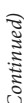

| Author [year]; country | Aim | Design; source of data | Population [sample size]; setting | Clinical disease [intervention] | Outcome(s) measured | Results | Prescribing determinants | Quality score |
|---|---|---|---|---|---|---|---|---|
| Tan et al., [2005]; Singapore [61] | Compare perceived factors influencing choice of anti-PD drugs with actual prescribing patterns. | Correlation study; patient records and neurologist survey. | Neurologists prescribing treatment to patients with PD [N=11 neurologists, N=306 patients]; tertiary hospital. | PD [anti-PD drugs]. | Factors influencing neurologists' choice of drug and prescribing patterns by drug type, class and combination. | L-dopa in combination with decarboxylase inhibitor was the most prescribed drug and most common for older patients in later stage PD. Cost and efficacy appeared to override other factors. | Age, disease severity, side effects, drug availability, clinical experience, drug cost, patient preference, drug company sponsorship. | 4* |
| Thomas et al., [2013]; UK [62] | Investigate the impact of medical agency warnings on antipsychotic prescribing for dementia. | Interrupted time series; locally developed electronic prescribing system. | Patients with dementia admitted to hospital [N=6,734 spells]; secondary care. | Dementia [antipsychotic prescribing]. | Changes in prescribing of clopidogrel and omeprazole since the introduction of new MHRA guidance. | Official medical agency warnings are associated with a reduction in antipsychotic prescribing. | Medical agency warnings. | 4* |
| Timotjevic et al., [2020]; Greece, Italy, Slovenia, UK [9] | Identify clinician needs to inform development of a CDSS for PD. | Mixed methods; Study 1: interviews/focus groups; Study 2: survey; Study 3: vignette. | Prescribing clinicians and nurses [Study 1 N=47; Study 2 N=12; Study 3 N=18]; primary and secondary care. | PD [CDSS]. | Types of decisions made, factors and combination of factors that contributed to decisions, judgement data under varying degrees of certainty. | Recommend using symptom data according to patient goals and desired QoL. Worsening symptoms predicted change of medication. | Worsening motor and non-motor symptoms, e.g., bradykinesia, tremor, cognitive function or impulsivity. Patient-directed goals, QoL, depression, age, employment, patient self-reported and objective data, consistent sources of data. | 4* |
| Trifirò et al., [2008]; Italy [63] | To evaluate the prevalence of use and prescribing pattern of anti-PD drugs in Italy. | Retrospective database analysis; GP prescribing database 2003–2005. | Users of anti-PD drugs [N= 1,479]; community. | PD [anti-PD drugs]. | Prevalence and incidence of patients prescribed anti-PD drugs, by drug type, class and combination. | L-dopa was the most prescribed drug. There was an increase in the use of ergot and non-ergot DAs, especially in older patients. | Age | 4* |
| van den Heuvel et al., [2022a]; Netherlands [64] | Clarify the process of personalised decision-making for PD treatment. | Qualitative; observational study and semi-structured interviews. | Audio recordings from N=19 clinicians; of those, n=16 took part in semi-structured interviews; outpatient clinics. | PD [personalised treatment]. | If, how and why decisions were personalised. | Clinicians balanced a range of clinical and non-clinical factors when making personalised decisions. Barriers of tailoring decisions was often due to difficulty in predicting effect in patients and lack of information on patient or disease variables. | Disease characteristics and symptom severity, patients' history, age, gender, personality, education level, adherence, cognitive ability, side effects, patient preference, self-management, psychosocial factors, QoL, evidence to support expected outcomes, type of clinician, guidelines, clinician experience, peer opinion, lack of clinical trials, MDT support. | 5* |

*(Continued)*

| Author [year]; country | Aim | Design; source of data | Population [sample size]; setting | Clinical disease [intervention] | Outcome(s) measured | Results | Prescribing determinants | Quality score |
|---|---|---|---|---|---|---|---|---|
| van den Heuvel et al., [2022b]; Netherlands [65] | Identify factors influencing, and likelihood of initiating, DMT for PD. | Quantitative; online survey. | Neurologists [N=64], neurology residents [N=18]; hospital. | PD [DMT]. | Decision choice based on treatment effect, risk of mild/severe side effects, route of administration, annual cost. | Clinicians were more likely to consider DMT treatment with patients at high risk of prodromal phases of PD. Treatment effect and risk of severe side effects were key influencing factors. | Stage of disease, side effects, evidence base, patient preference, life expectancy, adherence, monitoring, symptoms, lifestyle, comorbidity, cost effectiveness, family history of PD, partner support, insurance coverage. | 5* |
| Walker et al., [2018]; England [66] | Investigate prescribing trends for dementia drugs in England between 1997 and 2016. | Retrospective trend analysis; primary care database. | Patients with probable AD [N=10,651], possible AD [N=12,167], non-AD and mixed dementias [N=17,384]; primary care. | Dementia [AChE inhibitors and NMDA receptor antagonists]. | Average monthly prescribing changes for AChE inhibitors and NMDA receptor antagonists; trending searches for AD. | Prescriptions increased for AChE and NMDA drugs, likely in response to patent expiries and updated national guidance, respectively. Increased awareness of dementia may also play a role. | Patent expiry, national disease campaigns, national guidance, indication of drugs, progression and stage of disease. | 5* |
| Walsh et al., [2018]; Ireland [67] | Explore the determinants of antipsychotic prescribing for dementia. | Qualitative; semi-structured interviews. | HCPs, family members [N=27]; nursing home residences. | Dementia [antipsychotics]. | Domains of the TDF influencing prescribing behaviour. | Nine TDF domains were found to influence prescribing behaviour. Balancing the risk and benefits was a key overarching factor. | Risk versus benefit of prescribing, QoL, distress of staff and family members, pressure to prescribe, regulations, guidelines, staff training, knowledge, side effects, fear of negative consequences, care resource, clinician communication and interpersonal dynamics, e.g., perceived hierarchy. | 5* |
| Wei et al., [2015]; USA [68] | To examine nonadherence of antiparkinnson drugs (APD) and regimen changes and impact on healthcare use and costs. | Retrospective cohort analysis; Medicare claims data. | Patients with PD prescribed ≥2 APD [N=7,052]; in/out patient hospital setting. | Parkinsons [APD's]. | Medication adherence, regimen changes, healthcare use and costs. | Modifications were higher in patients non adherent and correlated to higher healthcare use and costs. | Non adherence | 5* |
| Werner et al., [2006]; Israel [69] | Assess family physicians' recommendations for AD treatments. | Experimental vignettes; phone survey. | Family physicians [N=395]; primary care. | AD [pharmacological and non-pharmacological treatment]. | Physician knowledge and attitudes of AD. | Physicians mostly recommended non-pharmacological options. Prescribed medication was associated with disease and changes in disease severity. | Severity of disease; patient perceived as dangerous. | 4* |

(Continued)

**Table 1.** (Continued)

| Author [year]; country | Aim | Design; source of data | Population [sample size]; setting | Clinical disease [intervention] | Outcome(s) measured | Results | Prescribing determinants | Quality score |
|---|---|---|---|---|---|---|---|---|
| Wood-Mitchell et al., [2008]; UK [70] | To explore factors that influence old age psychiatrists to prescribe for BPSD in dementia | Qualitative study; focus group and 1:1 interview data. | Old age consultant psychiatrists [N=8] | Dementia [medication for BPSD] | Factors explaining when and why psychotropic medication would be prescribed. | Inconsistent choice of medication prescribed. Common reasons for prescribing are reported in next column. | Pressure to prescribe, lack of resources, lack of alternative treatment, professional skills/experience, risk-benefit ratio of harm, familiarly of drug, side effect profile, evidence base, guidelines, peer influence, polypharmacy, knowledge of patient. | 4* |
| Zuidema et al., [2011]; Netherlands [71] | Examine if staff distress and nursing home environment are associated with PDU. | Observational cross sectional; residence medication record, observations of nursing home environment, survey data. | Nursing home residents with dementia [N=1,289], number of vocational nurses is not reported; care home. | Dementia [psychotropic drugs]. | Presence or absence of PDU. Nursing home environment and staff distress were also measured. | Staff distress due to resident agitation, nursing home overcrowding, limited staff was associated with PDU. | Staff distress due to resident agitation, nursing home environment and neuropsychiatric symptoms. | 5* |

AChE, acetylcholinesterase; AD, Alzheimer's disease; BPSD, behavioural and psychological symptoms of dementia; CDSS, Clinical Disease Support Service; ChEI, cholinesterase inhibitor; DA, dopamine agonist; DMD, disease-modifying drug; DMT, disease-modifying treatment; GP, general practitioner; HCP, healthcare professional; ICD-9, International Classification of Diseases, ninth revision; L-dopa, levodopa; MDT, multidisciplinary team; MHRA, Medicines and Healthcare products Regulatory Agency; MS, multiple sclerosis; NHS, National Health Service; NICE, National Institute for Health and Care Excellence; NMDA, N-methyl-D-aspartate; PD, Parkinson's disease; PDU, psychotropic drug use; QoL, quality of life; SIGN, Scottish Intercollegiate Guidelines Network; TDF, Theoretical Domains Framework.

efficacy and prescribing guidelines were also common factors, each reported by 15 studies. Factors least reported to influence prescribing include patient education, carer burden, family history, insurance, psychosocial factors, negative treatment experience, personality, logical fallacy, care/physician burden, HCP apathy, refusal to prescribe, staff distress, delivery of care, level of care, patent expiry, national campaigns and drug sponsorship.

A matrix grid was developed to categorise each factor into patient, HCP or healthcare system groups, and initially mapped to the TDF domains, then onto COM-B components to facilitate identifying key behavioural determinants (see Fig 2 for full overview). Most factors influencing prescribing centred around patients (n=24) and HCPs (n=20), with the healthcare system group (n=12) having the fewest. Theoretically, most factors were underpinned by opportunity (n=21), followed by capability (n=20) and motivation (n=15). From the TDF domains, patient-related factors were found to commonly influence HCP decision making, a domain which maps onto capability when applying the COM-B model. HCP-related factors mostly influenced TDF domains, such as belief about consequences, belief about capabilities, professional and social role, goals, reinforcement and emotion, which are linked to COM-B motivation. Factors related to the healthcare system group were primarily determined by the TDF domains social influence and environmental context and resource, which links onto COM-B opportunity. Healthcare system was the only group that did not have factors mapped to all three COM-B domains (i.e., no factors were found to influence motivation).

## Capability

**Capability – patient-related factors.** Most studies (n=40, 74%) reported patient-related factors. These factors commonly informed the decision process (mapping to capability) that HCPs worked through when identifying appropriate medication. For example, prescribing was described as, "… the lesser of two evils",[29] (p. 8) when seeking to manage symptoms and HCPs often considered the impact of prescribing *versus* not prescribing, in relation to potentially improving,

**Fig 2. Heatmap of studies, split by capability, opportunity and motivation, and by patient, HCP and healthcare system domains.** HCP, healthcare professional; MDT, multidisciplinary team.

halting or risking a worsening of symptom severity and burden, particularly when starting, changing or discontinuing medication [19,22,23,26–28,33,37,40,41,44,45,48,53,58,64–66]. HCPs frequently considered patient age when making prescribing decisions [22,24,26,31,40,42,46,48,51,53,58,60,61,63,64]. While age ranges reported were often inconsistent between studies,there was indication of variation in types of medication prescribed to patients </>70 years of age. The reason for such variation was often due to an associated risk of adverse events, for example,where there was a heightened risk of cognitive side effects, fractures or hospitalisation related to anticholinergics prescribed to patients with Parkinson's disease >70 years of age [48,51] and problems of levodopa (L-dopa) dyskinesia in younger patients [61]. Although patient gender was commonly explored as a potential determinant for prescribing, the evidence to support this was mixed [24,42,51]. For example, one study investigated gender differences associated with anticholinergic, dopamine agonists (DA), ergot DA, non-ergot DA, L-dopa and MAO-B inhibitor drugs and only found significance for anticholinergics; women were 32.2% more likely than men to receive a prescription. Conversely, another study [24] contradicts the former [51], whereby the authors found that physicians preferred to prescribe rasagiline (a MAO-B inhibitor) for men. Identifying the stage and trajectory of a disease state was also important [28,32,48,64,66] because it may influence the number of different drugs prescribed [32] or the drug required to reach new treatment goals, due to disease progression [23,28]. HCPs made decisions on whether or not a medication was suitable for patients, which commonly involved consideration of patient medication history, tolerability of a drug, route of administration, adherence and pill burden [28,32,33,37,53,54,64,66]. The context around these decisions was further complicated by comorbidity [23,24,28,33,36,42,53,72] and polypharmacy [23,24,26,28,36,46], particularly when medication was prescribed to manage symptoms of depression [26,30,51]. Lastly, patient hospital visits were perceived to influence prescribing [53,58], particularly when initiating [26,46] or discontinuing [33] a prescription,and patient level of education reportedly influenced patient and provider medication preference [66].

**Capability – HCP-related factors.** More than half of the studies (n=29, 54%) reported factors that mapped to HCP skill or knowledge (and subsequently capability). Of these, 18 studies reported that HCPs perceived drug safety or efficacy to be an important determinant of prescribing [32,33,36,37,39,43–46,52,54,55,58]. For example, one study [52] found that HCPs were more likely to prescribe a hypothetical new Alzheimer's drug if it maintained or delayed disease progression. Conversely, one study disputed that HCP perceptions of efficacy are influenced by knowledge or beliefs of treatment benefits [39], and identify a need for further education [69], and where it can improve prescribing practice [47]. HCP experience and familiarity of using a drug generated a greater sense of confidence (and objectively perceived confidence) in prescribing [20,30,36,39,43,47], meaning that with more experience came more confidence and better clinical judgement [30,43,66]. Confirming a diagnosis also influenced the likelihood of clinicians prescribing certain medication [30,41–43,45,53], whereas underdiagnosing [45] or no diagnosis [41] delayed treatment; one study reported that HCPs adjusted diagnosis to avoid delays, so that they could prescribe certain medication [43] clearly warranting further investigation into such behaviour. HCP current knowledge of drug interaction and properties, including route of administration [28,39,47], and ability to identify relapses, specifically for multiple sclerosis [20,37], may influence the quality of appropriate prescribing decisions [54].

**Capability – Healthcare system-related factors.** Several studies (n=21, 39%) were found to report factors linked to the healthcare system; these factors commonly influenced HCP knowledge and skill and were subsequently mapped onto the COM-B domain, capability. Within this domain, factors linked to prescribing guidelines were most often reported [20,30,36,41,44,66,67]. However, HCP views, knowledge of and adherence to guidelines were mixed; some studies reported HCPs who dismissed guidelines [30,44,57], followed them or agreed with some rather than others [36,41,66,67]. For example, the Scottish Intercollegiate Guidelines Network (SIGN) and the National Institute for Health and Care Excellence (NICE) guidance is often used interchangeably across the UK, yet SIGN guidance was more accepted than NICE among HCPs prescribing antidementia medication in Scotland [30]. Studies also identified a need for more guidelines [43] and referred to a lack of evidence [23,36,43,65,66], skilled training [21,23,43], staff [21] and resources

[21,43], all contributing factors to potentially inappropriate prescribing. Knowledge and evidence supporting the indication of a drug also influence choice and the potential to optimise prescribing [55,67].

## Opportunity

**Opportunity – patient-related factors.** Nearly half of the studies (n=21, 39%) reported factors related to patient social and physical opportunity that influenced HCP prescribing decisions. Social factors in this group included patient and carer treatment preferences [28,32–35,37,64–66] and were reported to influence shared decision making [28,60,65], patient/carer agency [33,34,57] and HCP drug choice [32,35,37,64,66]. In terms of physical opportunity, there was some evidence that patient locality [26,31,40] and socio-economic status [9,60] were contributing factors for being prescribed certain medication. For example, studies conducted across Europe and the USA reported regional differences in the likelihood of receiving certain prescriptions [26,31,40] and, in particular, areas of higher deprivation were linked to antipsychotic prescribing [60], and patient employment influenced changes in Parkinson's disease care plans [9]. Some studies also reported that HCPs established whether patients had skilled support at home to help manage medication regimes [36,56,72] and considered the impact of carer burden [23] before making prescribing decisions. Additionally, HCPs prescribing treatment for patients with Parkinson's disease in the Netherlands reported that psychosocial factors [64], family history and insurance were further factors influencing drug choice [26,72].

**Opportunity – HCP-related factors.** A small number of studies (n=13, 24%) reported factors mapped to HCP opportunity. Factors underpinning this behavioural determinant were related to social expectations of prescribing linked to a prescribing culture or network [20,29], where HCPs sought reassurance and justified a prescribing norm, or felt a pressure to prescribe [21,27,67] from colleagues and carers, often due to limited healthcare resources. For example, one study [21] reported how 91% of GPs felt influenced by nurses and 59% were influenced by patient family members to prescribe medication, though this percentage reduced with clinician experience. Medication reviews [19,23,28,41] were an important factor and fit here as an opportunity determinant because they require HCPs to follow up with patients [41] and consider recommendations from pharmacists about medication choices [19,23,28]. Lastly, the ease and availability of medication influenced prescribing [34,37,43] and were perceived to help manage care, particularly for patients with worsening symptoms of dementia [34,69].

**Opportunity – healthcare system-related factors.** Some of the studies (n=25, 46%) reported factors associated with the healthcare system and were mapped to social or physical opportunity. Prescribing was mostly influenced by physical opportunities (i.e., the physical ability to prescribe). For example, lack of resources [21,33,34,36,41,43,55,69], including insufficient staffing [21,34,60], access or feasibility of non-pharmacological treatment [21,43,54] and time to discuss medication options during clinical appointments [36]. Evidence on drug costs was mixed; some studies reported how it was not and should not be a barrier to prescribing [43,44]; however, there was still evidence linking drug choice and cost [33,45,72]. Some barriers to prescribing stemmed from policy or regulatory restrictions [23,26,39,45,55,67], such as formulary coverage [39] or HCP prescribing permissions [56]. Environmental settings [29,71], such as nursing home facilities, also influenced prescribing (e.g., available walking circuits and poor staff-to-resident ratio) [71]. Changes in patent expiry and national campaigns for dementia increased acetylcholinesterase prescriptions [66], and drug sponsorship was found to influence choice of drug for patients with Parkinson's disease in Spain [61]. Multidisciplinary team (MDT) support was the only factor that mapped to social opportunity. MDT included interprofessional collaboration with specialists, nurses and pharmacists and was considered to help optimise and influence prescribing decisions, particularly among HCPs based in community care [21,23,27,28,36,59,64].

## Motivation

**Motivation – patient-related factors.** Studies that reported patient-specific factors (n=24, 44%) were mapped to motivation. Patient-specific factors often influenced HCP evaluation of treatment consequences and treatment goals,

which were also impacted by HCP beliefs or intention toward a behaviour (i.e., reflective motivation). For example, the risk of patients experiencing side effects was reported most often as a patient-specific factor [23,28,32–35,41,43,55,60,64,66,69], and this risk commonly influenced treatment choice [28,32,64], as well as the decision to either switch [33] or discontinue medication [33,41]. Patients who reported a negative experience with medication also influenced prescribing because HCPs did not want to risk repeating another negative outcome [19,36] (one study links this to cognitive bias, outlined in the HCP paragraph below). Maintaining patient cognitive [9,23,33,53,58,64] and physical function [33,53,58] was important; a decline or change in either of these influenced patient and HCP treatment choice [9]; however, such changes are difficult to determine among patients with dementia [23]. Patient personality was a concern, particularly when HCPs believed there was a risk of triggering impulsive control disorders in patients with Parkinson's disease [9,22,64]. A decision to prescribe was more likely when HCPs believed medication would improve patient quality of life (QoL) [9,28,32,34,43,56,64,65,67] or if it was the only option in the absence of non-pharmacological alternatives [34]. Increasing the risk of reducing patient quality of life due to withdrawing medication influenced some HCPs to continue prescribing [21,27]. Treatment goals varied depending on disease stage [69], symptoms [23] and changes in prognosis [28], therefore, importance was placed on identifying patient treatment goals and prescribing accordingly [9,28,43].

**Motivation – HCP-related factors.** Several studies (n=24, 44%) identified HCP-related factors that were mapped to motivation. Factors mapped here were mostly underpinned by HCP professional identity, belief of their capability to treat patients and consequences of their decisions (i.e., reflective processes). For example, specialists were more likely to issue a prescription following a consultation with a patient [26,31,33,40,42,43,57,59], while GPs sought specialist support on what to prescribe [34] and also reserved any off-label prescribing to specialists [43]. HCP gender influenced prescribing [39,44,57], albeit to a lesser extent (e.g., females were more likely to initiate medication at lower doses [57], whereas in another study men were more likely to prescribe a new licenced drug [44]). Studies reported how HCPs believed that the benefits of prescribing should always outweigh the risks [23,28,29,34,36,41,43,69]; however, this was difficult to assess due to patient variability, comorbidities and polypharmacy [23,28,36,43]. Therefore, HCPs feeling confident in their own ability to make such judgements was an influencing factor, and where this was lacking it triggered more cautious prescribing [20,27,36]. There was further evidence that factors influencing motivation were also underpinned by emotion and to a lesser extent habit (automatic processes). For example, fear or distress influenced prescribing, particularly when HCPs were concerned about potential consequences of not prescribing or changing a medication and triggering a worse outcome [22,23,29,34,43,69]. Habitual prescribing was an influential factor and commonly driven by cognitive biases developed from HCP negative experiences of prescribing [36], prescribing inertia [28,36] or, more broadly, prescribing was described as the path of least resistance [36].

## Discussion

This rapid review included 54 studies, of which more than half were considered high quality (57%) and describes a range of factors that influence prescribing for NDs. The review had two objectives: 1) identify factors influencing prescribing for NDs and 2) map factors to theoretical determinants to better understand what underpins prescribing practices. A complex range of 56 factors was identified and grouped into factors specific to patients, HCPs or the healthcare system. Prescribing was mostly influenced by patient-related factors (specifically those associated with patient medical profile, e.g., stage and burden of disease symptoms) as well as individual HCP factors (mostly regarding HCP clinical experience and knowledge of drug profiles). The healthcare system also influenced prescribing, albeit to a lesser degree, with factors centred around the lack of resources or access to multidisciplinary support and guidelines. When mapping the factors influencing prescribing, a behavioural analysis using the TDF and COM-B model revealed that HCP prescribing practices are mostly determined by opportunity (e.g., HCP social interactions) and their own capability (e.g., their ability to retain and apply information to inform a prescribing decision). Motivation was also an important determinant and is likely influenced by

factors underpinned by HCP beliefs about consequences of prescribing or not prescribing and their own capabilities when manging medication treatment, as well as their professional and social role and their emotional responses to patients.

To the authors' knowledge, this review is the first to directly apply both the TDF and COM-B models together to identify and understand the behavioural determinants influencing prescribing, specifically for NDs. By understanding the drivers of prescribing in this way, it allows the review to comment on and add to knowledge about behavioural patterns that, if targeted, could help optimise prescribing. Efforts to optimise [36] or change [73] prescribing habits are widely reported, and there are already evidence assessing intervention effectiveness [11,74]. There is, however, consensus within the literature that calls for more theory-informed research to a) understand the behavioural determinants of prescribing and b) identify the mechanisms by which a change in prescribing may be triggered, both of which this review investigated [11,74].

The factors influencing prescribing identified in this review are consistent with factors reported in another review [75] that investigated the pattern and determinants of prescribing for Parkinson's disease. Both reviews identified patient age, gender, comorbidity and disease duration as influencing patient factors; however, Orayj and Lane only identified type of prescriber as an influencing HCP factor [75]. In contrast, this review refers to clinical specialism rather than type of prescriber and highlights a range of other HCP factors that frequently influenced prescribing (e.g., medication reviews, prescribing culture, ease of prescribing and staff distress). Furthermore, this review also highlighted several healthcare system-related factors as important influences on prescribing, including a lack of clear guidelines and adherence to guidelines [36,59,67]. While the current work focused on Parkinson's disease, future research should consider different diseases, such as Alzheimer's or Huntington's disease, as well as the extent of behaviour change that is driven by the type of prescriber treating these patients.

Understanding the competing cognitive demands associated with prescribing may be key for making optimal decisions [9]. This review highlights that the capability of HCPs (i.e., their knowledge and skills) is an important determinant in prescribing. For example, understanding the relevant optimal range of drugs to prescribe for a patient is one of many cognitive actions that HCPs simultaneously balance and is often processed by establishing and evaluating a range of patient-specific information [9]. Evidence to improve this clinical decision making suggests that techniques such as information framing (e.g., using a credible source to communicate information on the originators behalf), changing default prescribing options or enabling choice effectively improves clinical decisions [76]. The role of HCP capability as a determinant for optimised prescribing is further supported by a systematic review that found 16 (80%) interventions effective in targeting optimised prescribing [11]; the review highlights that the TDF domain 'memory, attention and decision processes' (which maps onto capability) was the most common behavioural determinant and, to target this, the behaviour change technique 'prompt/cues' was often used. Additionally, the studies included in the review by Talat et al. found that social opportunities (e.g., social influences and environmental context/resources) were common behavioural determinants relevant to suboptimal prescribing [11]. Key social influences identified in our review include patient and carer preferences and support to manage medication regimes [23], prescribing cultures [77] or availability of healthcare resources or skilled staff [21,77]. Behaviour change interventions targeting prescribing behaviour have shown that social reference point strategies are very effective in changing physician behaviour, and applying techniques such as prescribing feedback, peer comparison and sharing social norms may be useful to overcome opportunity barriers [74]. Techniques listed under the identity cluster in the behaviour change taxonomy may be relevant to target the social influences of medication prescribing [11]. In our review, prescribing decisions were clearly influenced by HCP motivation and characterised primarily by their professional role or identity, belief about their own capability, and possible prescribing consequences and emotional responses. These behavioural determinants were also used by several of the interventions identified by Talat et al. [11], although it should be noted that professional role or identity was not often targeted due to a lack of appropriate techniques.

The findings of this review enhance our understanding of the wide range of behavioural determinants that underpin prescribing for NDs. Such insight enables the identification of behaviour change techniques that are theoretically best suited and likely to be more effective in targeting optimal prescribing. The findings reported here can be applied to the

next stages of the Behaviour Change Wheel framework that supports intervention designers to select relevant intervention content. Our findings also enable future quantitative work to identify which of these factors are the strongest statistical predictors of prescribing behaviour for NDs,which will also contribute to interventions designed to optimise this. A strength of this review is that it offers a comprehensive and up-to-date summary of factors influencing prescribing for NDs and enables a theoretical understanding of the behavioural determinants underpinning prescribing practice. The following limitations should also be recognised. First, key search terms did not include any common ND medication names (e.g., L-dopa or donepezil) or drug classes (e.g., cholinesterase inhibitors or DAs), which may have identified further relevant articles. Second, many factors influencing prescribing were patient specific (e.g., age, gender, symptoms or comorbidities) which the TDF and COM-B model does not easily lend itself to. As such, these factors were mapped onto the TDF domain 'memory, attention and decision making' because they informed the HCP decision-making process. Third, the review only included articles published in the English language and available open access; this was to facilitate the rapid process of the review but, as such, the importance of studies reported in other languages should be considered in future work.

This review provides an overview of prescribing behaviour across a number of NDs and highlights a number of potential determinants of suboptimal prescribing. In order to optimise prescribing, this review could be used, alongside a review or audit of prescribing for a condition or a prescribing audit for a specific medication or class of medications, to understand the factors that might be driving suboptimal prescribing of a specific medicine/class of medicines/for a specific ND (e.g., Parkinson's disease).

As the population ages, the need to optimise medication profiles has increasingly been discussed in the healthcare literature. In conclusion therefore, examination of the literature here reveals that prescribing for NDs is influenced by a range of factors that are underpinned by the following behavioural determinants: 1) HCP capability, specifically factors influencing their decision-making processes (e.g., patient ages and symptoms) and clinical knowledge; 2) HCP opportunity, including access to MDT support and medication availability; and 3) HCP motivation, in regard to beliefs about worsening side effects or symptoms and perception of their own specialism. Interventions that aim to optimise medication choices for NDs should consider the appropriateness and acceptability of targeting these determinants and identify relevant intervention functions and behaviour change techniques to increase the likelihood of change. The transferability of these factors and determinants in relation to prescribing for other disease areas should also be explored with a view to improving prescribing practices more generally.

## Supporting information

**S1 Appendix.  Full inclusion and exclusion criteria.**
(DOCX)

**S2 Appendix.  Full search strategy for Ovid MEDLINE.**
(DOCX)

**S3 Appendix.  TDF codebook.**
(DOCX)

**S4 Appendix.  Quality assessment using the Mixed Methods Appraisal tool.**
(DOCX)

**S5 File.  Statement on industrial partnership.**
(DOCX)

**S6 Appendix.  Excel file of Excluded articles.**
(XLSX)

## Acknowledgements

Giada Zanella is a library and information specialist at Aston University who supported the development of the search strategy. William Hind and Dr Catrina Milgate from Alpharmaxim contributed to reviewing and editing draft versions of the manuscript; they also oversaw the supervision of resource support from professional medical writers and editorial assistants who data checked and formatted the manuscript. Dr Rebekah Young, also from Alpharmaxim, supported with literature reviewing and the data extraction process. We would like to thank the two reviewers for this paper who provided constructive and supportive comments that contributed to an improved article. We would specifically like to thank Dr Hannah Family whose comments on the subsequent optimisation of prescription behaviour have been incorporated into the discussion of the report.

## Author contributions

**Conceptualization:** Jason Thomas, Carl Senior.

**Data curation:** Emma Begley, Jason Thomas.

**Formal analysis:** Emma Begley, Jason Thomas.

**Funding acquisition:** Jason Thomas, Carl Senior.

**Investigation:** Emma Begley, Jason Thomas, Carl Senior.

**Methodology:** Emma Begley, Jason Thomas.

**Project administration:** Emma Begley.

**Resources:** Emma Begley, Jason Thomas, Carl Senior.

**Software:** Emma Begley, Jason Thomas, Carl Senior.

**Supervision:** Jason Thomas, Carl Senior.

**Visualization:** Emma Begley.

**Writing – original draft:** Emma Begley.

**Writing – review & editing:** Emma Begley, Jason Thomas, Carl Senior.

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
