## [Decision Letter · Decision Letter 0]

28 Oct 2024

PONE-D-24-24221A behavioural analysis of factors influencing prescribing for neurodegenerative diseases: a rapid reviewPLOS ONE

Dear Dr. Senior,

Thank you for submitting your manuscript to PLOS ONE. After careful consideration, we feel that it has merit but does not fully meet PLOS ONE’s publication criteria as it currently stands. Therefore, we invite you to submit a revised version of the manuscript that addresses the points raised during the review process.

We look forward to receiving your revised manuscript.

Kind regards,

Vaibhavi Peshattiwar, PhD

Guest Editor

PLOS ONE

Journal Requirements:

“I have read the journals policy and the authors of the manuscript have the following competing interests: the research is part funded by Alpharmaxim Healthcare Communications Limited, where EB has undertaken the KTP-associate placement and therefore has a professional interest. CS and JT are both employed by Aston University and have no competing interests.”

3. We note that you have included the phrase “data not clearly reported” in your manuscript. Unfortunately, this does not meet our data sharing requirements. PLOS does not permit references to inaccessible data. We require that authors provide all relevant data within the paper, Supporting Information files, or in an acceptable, public repository. Please add a citation to support this phrase or upload the data that corresponds with these findings to a stable repository (such as Figshare or Dryad) and provide and URLs, DOIs, or accession numbers that may be used to access these data. Or, if the data are not a core part of the research being presented in your study, we ask that you remove the phrase that refers to these data.

4. As required by our policy on Data Availability, please ensure your manuscript or supplementary information includes the following: 

5. Please provide with the supporting information about the role of role of Alpharmaxim Healthcare Communications Ltd in the research.

7. Please update the references with their respective doi. The reference guidelines can be found at https://journals.plos.org/plosone/s/submission-guidelines#loc-references

Reviewers' comments:

Reviewer's Responses to Questions

**Comments to the Author**

1. Is the manuscript technically sound, and do the data support the conclusions?

Reviewer #1: Yes

Reviewer #2: Yes

2. Has the statistical analysis been performed appropriately and rigorously? 

Reviewer #1: N/A

Reviewer #2: N/A

3. Have the authors made all data underlying the findings in their manuscript fully available?

Reviewer #1: Yes

Reviewer #2: Yes

4. Is the manuscript presented in an intelligible fashion and written in standard English?

Reviewer #1: Yes

Reviewer #2: Yes

5. Review Comments to the Author

Reviewer #1: Thank you for inviting me to review this rapid review of the literature which has been capably and comprehensively conducted. I applaud the authors for their work and diligence in adhering to all the reporting guidelines and appropriate checklists. It is also pleasing to see a mix of inductive and deductive approaches being applied to the analysis as this is something that has been raised as a potential limitation of applying the COM-B/TDF to the mapping of data if this is done without exploration of other factors. I have a few minor issues to be addressed.

1. What was the role of Alpharmaxim Healthcare Communications Ltd in the research? The acknowledgement section recognises the contributions but I wondered why they were involved in the first place? Are there any potential conflicts? It is difficult to judge this without further supporting information.

2. Introduction - I was wondering why all NDs were pooled for the purpose of this review. This became clearer later on in the Methods section when it is explained that preliminary searches for PD only yielded few outputs. However the reader is left wondering about this much earlier in the paper. Can the rationale for this be made clearer earlier on?

3. Results - The final number of PD papers was 13 which is sufficient for a rapid review. Whilst I liked the way the data were synthesised - I would have liked to have seen some categorisation of data for specific NDs such as PD and Alzheimer's. I feel that this would be more beneficial for clinicians in practice and would strengthen the significance / implications of the findings. Are there plans to do this in the future? I am not suggesting a reanalysis at this stage, but can these issues be addressed somewhere in the Discussion?

Line 198 - what does 'logical fallacy' mean? I think I can guess, but it is not a term that I am familiar with.

Line 217 and others - please add 'Capability - Patient related factors' to the sub-heading for Patient, HCP and healthcare system factors etc. for all the COM - otherwise it it easy for the reader to loose track of which COM-B domain is referred to.

Line 260 - I was shocked to read that HCPs adjusted the diagnosis to avoid delays in prescribing (ref 41) - this should be picked up for Discussion and merits further investigation.

Line 333-334 - I was not sure what the sentence in parenthesis meant? Is this needed?

Line 341 - add 'reducing' before QoL.

Line 361 - trigging - should this be 'triggering'?

Line 391-391 - I was unsure how Ref 69 had been managed during the searching and screening process (systematic review). Were the individual studies included in the 48? Please make this clearer.

Reviewer #2: It was a pleasure to read and review this paper reporting a rapid review of the behavioural determinants of prescribing for neurodegenerative diseases. Overall, this is a well conducted and well reported review that I enjoyed reading. I think it will be a useful contribution to the field - and particularly useful for any teams who carry out audits of prescribing practices. and, I have a few minor comments, to aid those less familiar with behaviour change theory, I hope these are helpful.

Abstract:

- can you indicate in here the type of studies included in the review, and also stating that it is a rapid review in your abstract would be helpful

- your conclusion could be improved I think. You have taken the time in the discussion, to consider potential behaviour change techniques that could address behavioural determinants of prescribing, and help optimise prescribing. You don't have space to list them all, but perhaps including one example in the conclusion of your abstract would be useful, and I think it creates a nice 'hook' to keep people reading onto the main body of the paper.

Methods

I think somewhere in your methods, probably the study inclusion criteria, to be explicit that you were including all types of prescribing. Although I do wonder whether your search strategy really captures all types of prescribing as there were no search terms for non-medical prescribing / supplementary or independent prescribing, You state in the discussion that the review was focussed on clinical specialism rather than type of prescriber but - I am not entirely sure what you mean by this, and whether you have considered the different types of prescribers that people with ND might encounter. So I think it would be very useful to be clear in the methods, whether your review was focussed on medical prescribing only, or whether you sought to capture non-medical prescribing in this review too.

Discussion

Where you discuss behaviour change techniques in relation to your COM-B/TDF analysis, I think it would be helpful to add in an explainer, (for those not used to behavioural science) along these lines: "Our review provides an overview of prescribing behaviour across a number of neurodegenerative conditions, and highlights a number of potential determinants of sub-optimal prescribing. In order to optimise prescribing, this review could be used, alongside a review or audit of prescribing for a condition or a prescribing audit for a specific medication or class of medications, to understand the factors that might be driving sub-optimal prescribing of a specific medicine / class of medicines / for a specific neurodegenerative condition (e.g. PD)." --- At the moment, your review covers so many types of prescribing behaviour and different conditions, and prescribing can be sub-optimal in so many different ways (e.g. prescribed too much, wrong formulation, wrong dose / suboptimal dose, wrong dosing schedule, not prescribing to guidelines, not prescribing within local prescribing budget limits, co-prescribing with other drugs when its contraindicated etc), it isn't really a behavioural analysis, as you haven't specified your target behaviour. Narrowing down the target behaviour, would slim down the number of behavioural determinants you had to work with / design an intervention to address.

6. PLOS authors have the option to publish the peer review history of their article (what does this mean? ). If published, this will include your full peer review and any attached files.

**Do you want your identity to be public for this peer review?** For information about this choice, including consent withdrawal, please see our Privacy Policy .

Reviewer #1: **Yes: ** Prof Delyth James

Reviewer #2: **Yes: ** Dr Hannah Family

---

## [Author Response · Author response to Decision Letter 1]

12 Feb 2025

AUTHORS’ RESPONSE

We have taken this opportunity to review the manuscript and have ensured that all formatting meets the journal style requirements.

“I have read the journals policy and the authors of the manuscript have the following competing interests: the research is part funded by Alpharmaxim Healthcare Communications Limited, where EB has undertaken the KTP-associate placement and therefore has a professional interest. CS and JT are both employed by Aston University and have no competing interests.”

As requested, we have included the following statement in the cover letter and we are grateful that you will update the online submission form on our behalf:

‘This does not alter our adherence to PLOS ONE policies on sharing data and materials.’

3. We note that you have included the phrase “data not clearly reported” in your manuscript. Unfortunately, this does not meet our data sharing requirements. PLOS does not permit references to inaccessible data. We require that authors provide all relevant data within the paper, Supporting Information files, or in an acceptable, public repository. Please add a citation to support this phrase or upload the data that corresponds with these findings to a stable repository (such as Figshare or Dryad) and provide and URLs, DOIs, or accession numbers that may be used to access these data. Or, if the data are not a core part of the research being presented in your study, we ask that you remove the phrase that refers to these data.

We have revised the sentence where this phrase occurs, as suggested (l.182).

4. As required by our policy on Data Availability, please ensure your manuscript or supplementary information includes the following:

A numbered excel file table of all studies identified in the literature search, including those that were excluded from the analyses, has now been included as supplementary attachment, S6.

The full inclusion and exclusion criteria for all studies are detailed in S1. Additionally, we have provided a table listing all studies identified through the literature search, including those that were excluded (see S6 and response above). Based on your suggestions, we identified five additional articles that met our inclusion criteria. These citations have been incorporated into the manuscript, with updated citation counts and references throughout. We have also revised the inserted table, descriptive narrative, Fig. 1 (PRISMA flow diagram), Fig. 2 (heatmap), and S4_file (both Marked-up and FINAL versions) to reflect the inclusion of these five studies. All updated files are attached.

No unpublished studies were included in this analysis.

Tables S1 and S2 contain details of the search terms and exclusion criteria for Ovid MEDLINE. Taken together, this provides sufficient detail to ensure that readers can replicate the study.

We confirm here that no data were used from these sources, only those sources that are indicated in the various tables noted above.

Thank you for highlighting this critical point for our readership regarding the assessment of our findings' validity. Our methodology adhered to the Cochrane Rapid Reviews Methods, emphasising comprehensive stakeholder involvement to refine the research question and eligibility criteria. Key steps included limiting interventions and comparators, prioritising significant outcomes and using justified restrictions (e.g., language and study designs). Searching was conducted with specialist involvement, focusing on core databases like Embase and Ovid MEDLINE, with peer-reviewed strategies. Screening and data extraction involved standardised forms, pilot exercises and dual reviewer systems. We used the EPOC checklist and MMAT for study quality assessment, with protocol registration on Prospero (See S3 and S4 for detail).

An explanation of how missing data were handled. This information can be included in the main text, supplementary information, or relevant data repository. Please note that providing these underlying data is a requirement for publication in this journal, and if these data are not provided your manuscript might be rejected.

As indicated above and validated through the MMAT assessment – no study was included that contained missing data.

5. Please provide with the supporting information about the role of role of Alpharmaxim Healthcare Communications Ltd in the research.

The industry partner for this Knowledge Transfer Partnership (KTP) project, in which this study was conducted, is Alpharmaxim. KTPs are UK government-funded initiatives that foster collaboration between academia and industry to explore specific research questions. This type of partnership leverages expertise from both sectors, providing clear benefits through collaborative research. As requested, we have ensured that a further statement regarding the role of Alpharmaxim is now uploaded as supporting information.

We would also like to highlight an additional change in the current affiliation of the first author of this article who is now employed as the Senior Public Health Practitioner in public health at the Liverpool City Council, UK.

We have taken the opportunity to review the reference list and confirm that there are no retracted articles. We also took the opportunity to address a handful of small typos that we identified.

7. Please update the references with their respective doi. The reference guidelines can be found at https://journals.plos.org/plosone/s/submission-guidelines#loc-references

We have ensured that DOIs are included for all references as requested.

Reviewers' comments:

Reviewer's Responses to Questions

Comments to the Author

Reviewer #1: Thank you for inviting me to review this rapid review of the literature which has been capably and comprehensively conducted. I applaud the authors for their work and diligence in adhering to all the reporting guidelines and appropriate checklists. It is also pleasing to see a mix of inductive and deductive approaches being applied to the analysis as this is something that has been raised as a potential limitation of applying the COM-B/TDF to the mapping of data if this is done without exploration of other factors. I have a few minor issues to be addressed.

Thank you very much for your thoughtful consideration of our work and the valuable points you have raised. We are delighted that you viewed our report so positively and we are confident that, together with the insights from R2, these suggestions have strengthened our report, making it of great interest to the journal’s readership.

1. What was the role of Alpharmaxim Healthcare Communications Ltd in the research? The acknowledgement section recognises the contributions but I wondered why they were involved in the first place? Are there any potential conflicts? It is difficult to judge this without further supporting information.

Thank you for raising this important point, which was also noted by the editor. While we have thoroughly addressed this in our previous response, we would like to further emphasise the pivotal role of the industry partner in our research. Conducted under a Knowledge Transfer Partnership (KTP) project framework, this collaboration merged industry and academia to solve significant research issues with mutual benefits. KTP guidelines ensured transparent agreements on roles, responsibilities and ethics from the start. Both parties worked towards shared research goals, supported by a governance structure with periodic management meetings and established communication channels. Agreements on data ownership, intellectual property and publication rights reinforced objectivity and trust. A supplementary statement detailing the industry partner's role has been provided for further clarity.

2. Introduction - I was wondering why all NDs were pooled for the purpose of this review. This became clearer later on in the Methods section when it is explained that preliminary searches for PD only yielded few outputs. However the reader is left wondering about this much earlier in the paper. Can the rationale for this be made clearer earlier on?

We have inserted the following statement (l.85):

‘As is noted below, the initial aim was to focus specifically on Parkinson’s disease, but this revealed limited citations, thus the search was subsequently broadened to include all NDs.’

3. Results - The final number of PD papers was 13 which is sufficient for a rapid review. Whilst I liked the way the data were synthesised - I would have liked to have seen some categorisation of data for specific NDs such as PD and Alzheimer's. I feel that this would be more beneficial for clinicians in practice and would strengthen the significance / implications of the findings. Are there plans to do this in the future? I am not suggesting a reanalysis at this stage, but can these issues be addressed somewhere in the Discussion?

You have highlighted an important point. It is worth noting that this research is part of a larger programme designed to examine a broader spectrum of specific NDs. As stated on l. 458, a key objective of the current paper is to demonstrate the transferability of these findings across various ND states. This transferability is intended to ensure that clinicians can readily adapt the recommendations derived from our evidence base to other NDs. By doing so, we hope these insights will offer practical, evidence-based guidance for managing a wide range of neurodegenerative conditions. To ensure the readership will benefit from your point, we have included the following statement in the discussion (l. 399):

‘While the current work focused on Parkinson’s disease, future research should consider different diseases, such as Alzheimer’s or Huntington’s disease…

Line 198 - what does 'logical fallacy' mean? I think I can guess, but it is not a term that I am familiar with.

The term describes an argument that may appear convincing or true but is fundamentally flawed. In the context of clinical decision-making, it refers to various cognitive biases, such as the bandwagon fallacy, where a clinician may prescribe a medication simply because it is popular among peers, or the false equivalence fallacy, where a clinician wrongly assumes two drugs are equally effective because they are both classified as neuroleptics, overlooking important differences in their mechanisms, side effects and efficacy. These cognitive shortcuts can lead to oversimplified decisions that overlook critical nuances in patient care.

Line 217 and others - please add 'Capability - Patient related factors' to the sub-heading for Patient, HCP and healthcare system factors etc. for all the COM - otherwise it it easy for the reader to loose track of which COM-B domain is referred to.

Done l. 216

Line 260 - I was shocked to read that HCPs adjusted the diagnosis to avoid delays in prescribing (ref 41) - this should be picked up for Discussion and merits further investigation.

This is indeed a surprising finding. We have inserted the following sentence at the point of the citation to highlight its importance (l. 261):

‘…clearly warranting further investigation into such behaviour.’

Line 333-334 - I was not sure what the sentence in parenthesis meant? Is this needed?

This sentence has been removed.

Line 341 - add 'reducing' before QoL.

Revised so the phrase now reads (l. 340:

…increasing the risk of reducing patient quality of life…’

Line 360 - trigging - should this be 'triggering'?

Apologies – this is a typo and has been corrected (l.360).

Line 391-391 - I was unsure how Ref 69 had been managed during the searching and screening process (systematic review). Were the individual studies included in the 48? Please make this clearer.

As is noted in l. 109 (and again in S1) any systematic reviews were screened for secondary references only.

Reviewer #2:

It was a pleasure to read and review this paper reporting a rapid review of the behavioural determinants of prescribing for neurodegenerative diseases. Overall, this is a well conducted and well reported review that I enjoyed reading. I think it will be a useful contribution to the field - and particularly useful for any teams who carry out audits of prescribing practices. and, I have a few minor comments, to aid those less familiar with behaviour change theory, I hope these are helpful.

Thank you very much for your support of our study. Your comments, along with those from R1, have significantly strengthened the article, and we are confident you will agree it will be of great interest to the journal's readership. We are also delighted to hear that you enjoyed reading our work. We have addressed each of the points you raised and provide a detailed, point-by-point response below.

Abstract:

- can you indicate in here the type of studies included in the review, and also stating that it is a rapid review in your abstract would be helpful - your conclusion could be improved I think. You have taken the time in the discussion, to consider poten

---

## [Decision Letter · Decision Letter 1]

19 Mar 2025

A behavioural analysis of factors influencing prescribing for neurodegenerative diseases: a rapid review

PONE-D-24-24221R1

Dear Dr. Senior,

We’re pleased to inform you that your manuscript has been judged scientifically suitable for publication and will be formally accepted for publication once it meets all outstanding technical requirements.

Kind regards,

Vaibhavi Peshattiwar, PhD

Guest Editor

PLOS ONE

Reviewer's Responses to Questions

**Comments to the Author**

1. If the authors have adequately addressed your comments raised in a previous round of review and you feel that this manuscript is now acceptable for publication, you may indicate that here to bypass the “Comments to the Author” section, enter your conflict of interest statement in the “Confidential to Editor” section, and submit your "Accept" recommendation.

Reviewer #1: All comments have been addressed

Reviewer #2: All comments have been addressed

2. Is the manuscript technically sound, and do the data support the conclusions?

Reviewer #1: Yes

Reviewer #2: Yes

3. Has the statistical analysis been performed appropriately and rigorously? 

Reviewer #1: N/A

Reviewer #2: N/A

4. Have the authors made all data underlying the findings in their manuscript fully available?

Reviewer #1: Yes

Reviewer #2: Yes

5. Is the manuscript presented in an intelligible fashion and written in standard English?

Reviewer #1: Yes

Reviewer #2: Yes

6. Review Comments to the Author

Reviewer #1: All my comments have been addressed. Thanks for the detailed responses. I wish you the best of luck with the larger programme of research within which this study sits.

Reviewer #2: (No Response)

7. PLOS authors have the option to publish the peer review history of their article (what does this mean? ). If published, this will include your full peer review and any attached files.

**Do you want your identity to be public for this peer review?** For information about this choice, including consent withdrawal, please see our Privacy Policy .

Reviewer #1: **Yes: ** Professor Delyth James

Reviewer #2: **Yes: ** Hannah Family

---

## [Editor Report · Acceptance letter]

PONE-D-24-24221R1

PLOS ONE

Dear Dr. Senior,

I'm pleased to inform you that your manuscript has been deemed suitable for publication in PLOS ONE. Congratulations! Your manuscript is now being handed over to our production team.

Kind regards,

on behalf of

Dr. Vaibhavi Peshattiwar

Guest Editor

PLOS ONE